# Application of Single-Cell RNA Sequencing in Ovarian Development

**DOI:** 10.3390/biom13010047

**Published:** 2022-12-27

**Authors:** Xiaoqin Gong, Yan Zhang, Jihui Ai, Kezhen Li

**Affiliations:** Department of Obstetrics and Gynecology, Tongji Hospital, Tongji Medical College, Huazhong University of Science and Technology, Wuhan 430030, China

**Keywords:** single-cell RNA sequencing, ovary, ovarian development, intercellular interaction

## Abstract

The ovary is a female reproductive organ that plays a key role in fertility and the maintenance of endocrine homeostasis, which is of great importance to women’s health. It is characterized by a high heterogeneity, with different cellular subpopulations primarily containing oocytes, granulosa cells, stromal cells, endothelial cells, vascular smooth muscle cells, and diverse immune cell types. Each has unique and important functions. From the fetal period to old age, the ovary experiences continuous structural and functional changes, with the gene expression of each cell type undergoing dramatic changes. In addition, ovarian development strongly relies on the communication between germ and somatic cells. Compared to traditional bulk RNA sequencing techniques, the single-cell RNA sequencing (scRNA-seq) approach has substantial advantages in analyzing individual cells within an ever-changing and complicated tissue, classifying them into cell types, characterizing single cells, delineating the cellular developmental trajectory, and studying cell-to-cell interactions. In this review, we present single-cell transcriptome mapping of the ovary, summarize the characteristics of the important constituent cells of the ovary and the critical cellular developmental processes, and describe key signaling pathways for cell-to-cell communication in the ovary, as revealed by scRNA-seq. This review will undoubtedly improve our understanding of the characteristics of ovarian cells and development, thus enabling the identification of novel therapeutic targets for ovarian-related diseases.

## 1. Introduction

The ovary is the primary female reproductive organ and can generate fertilizable oocytes to sustain female reproductive ability and secrete steroid sex hormones to regulate endocrine function [1]. The ovarian tissue structure is complex and consists of various types of cells, such as germ and somatic cells, containing oocytes, granulosa cells, stromal cells, vascular smooth muscle cells, endothelial cells, and immune cells [2,3,4]. Since each cell type and cell-to-cell interactions are essential for ovarian growth, development, and functional maintenance [5,6,7,8,9,10], it is of great significance to understand molecular mechanism changes in each cell type at various developmental stages and the communication between cells for the research and treatment of ovarian-related diseases. However, the characteristics of each cell type in the ovary are not well understood. To date, most research on the ovary has been conducted on cell populations [11,12,13,14]. Hence, it will be highly valuable to clarify the composition of ovarian cells, the characteristics of each cell type, and the interactions between cells at the single-cell level.

Over the last several decades, RNA sequencing technology has undergone rapid development and become a favorable method for transcriptome profiling [15]. It is an attractive tool for detecting changes in cellular gene expression and dissecting complex biological processes [16,17]. RNA sequencing has deepened our understanding of gene functions. The early RNA sequencing method was bulk RNA sequencing, which can examine the average gene expression in a large population of cells. However, it is difficult to accurately reveal cell-type-specific changes in gene expression, particularly in rare cell types [18,19]. In addition, it fails to characterize cellular heterogeneity [20].

Single-cell RNA sequencing (scRNA-seq) began to emerge since Tang et al. first analyzed the transcriptome of mouse oocytes and embryos at the single-cell level [21]. In recent years, this technology has developed rapidly and has been applied in many fields, such as neuroscience, immunology, cardiology, and oncology [22,23,24]. In contrast to conventional bulk RNA sequencing methods, scRNA-seq can comprehensively reveal the gene expression among individual cells, thus identifying distinct cell types as well as their cell-type-specific gene signatures, and can discover previously unknown cell types and subtypes, which might be beneficial to our understanding of rare cells and the functional roles of cells, especially in organs and tissues that comprise multiple cell types, such as ovaries, tumors, lungs, and the liver [23,25,26,27,28,29]. In addition, the cellular developmental trajectory can be mapped using scRNA-seq [30], thus contributing to a better understanding of the basic processes that occur during ovarian development. Additionally, scRNA-seq has the advantage of elaborating intercellular signaling pathways [31]. Considering that ovarian development is highly regulated by cell–cell interactions [32], applying scRNA-seq to ovarian research will also provide us with an unprecedented opportunity to resolve the interplay and exchange of communication between ovarian cells.

Currently, scRNA-seq has been widely used in ovarian studies and has provided novel ideas and made significant contributions to the further study of ovarian development and ovarian-related diseases. However, these findings were dispersed and not systematically conclusive. In this review, we focused on recent developments in scRNA-seq applied to the ovaries of humans, nonhuman primates, and mice to depict the cellular atlas and introduce the key physiological development processes and interactions of the important constituent cells of the ovary, which will facilitate the understanding of ovarian cellular heterogeneity and development, thus increasing the knowledge of ovarian-related diseases.

## 2. Ovarian Cell Heterogeneity at A Single-Cell Resolution

The cellular heterogeneity within the ovary poses a challenge to understanding its complex functions. Studying populations at a single-cell resolution has provided a complete atlas of the ovary and enabled the identification of cell-type-specific genes of ovarian germ and somatic cells, which may provide a new method for studying ovarian development.

### 2.1. The Single-Cell Atlas of the Ovary

Serving as an important female reproductive organ, the ovary contains many heterogeneous cell populations. However, little is known about the precise cell components that constitute the ovary. To solve this issue, scRNA-seq is a powerful method that can not only identify distinct phenotypic cell types, but also distinguish cell subpopulations, even in seemingly homogeneous cells, by single-cell gene expression [33].

By sequencing more than 24,000 cells from high-quality ovarian cortex samples from five cesarean section patients (20–38 years old) and sixteen sex reassignment patients (28–37 years old), Magdalena et al. [34] demonstrated that the human ovary comprised six main cell types: oocytes, granulosa cells, immune cells, endothelial cells, perivascular cells, and stromal cells. Furthermore, they found that the ovarian cortex of sex reassignment patients showed the same cell type as that of cesarean section patients, suggesting that androgen therapy in sex reassignment surgery patients does not affect ovarian cell types. In addition, the somatic cell types from the ovarian cortex were consistent with single-cell transcriptome profiles of tissue fragments from the inner part of the human ovary, including granulosa, immune, perivascular (or smooth muscle), endothelial, and stromal cells [35].

Furthermore, scRNA-seq research found that the ovarian cell types of mice were mostly in accordance with those of adult humans and included germ, granulosa, immune, endothelial, and stromal cells; however, scRNA-seq technology did not find smooth muscle cell clusters in the ovaries of fetal mice, ranging from embryonic day 16.5 (E16.5) to postnatal day 3 (PD3) [36,37,38]. Similarly, scRNA-seq analysis did not reveal smooth muscle cell clusters in the ovaries of human embryos between 5 and 26 weeks postfertilization [39]. The first smooth-muscle-myosin (SMM)-positive intraovarian cells in rats were discovered on PD6 by immunohistochemical staining [40]. SMM is a key regulator of smooth muscle contraction and a cell-specific marker of smooth muscle cells [41,42]. Previous studies reported smooth muscle cells in the ovaries of mice at PD21–PD23 and in rats between 3 weeks and 4 months [43,44], suggesting that the ovarian cell type may vary from the fetal to adult stages. Juvenile (4–5 years old) and aged (18–20 years old) cynomolgus monkeys showed the same ovarian cell types, and aged ovaries had fewer developmental follicles than the juvenile ovaries had [45], implying that the proportion of the ovarian cell population changes with age; however, the ovarian cell type showed no alterations in adult nonhuman primates. Additionally, the nonhuman primate ovary shares similar cell types with the human ovary [34,45], providing powerful evidence for choosing to use nonhuman primates instead of humans in some ovarian studies because of the inaccessibility of human tissues and related ethical constraints. Future comparisons of single-cell transcriptomic data of nonhuman primate ovaries with those of human ovaries will help discover similar genetic and physiological characteristics of nonhuman primates and human ovaries, which can provide profound insights into the reason for choosing primates in some research. Cross-species analyses may yield unique and similar insights into our understanding of ovarian cell diversity. Collectively, these defined subtypes deepen our understanding of the ovary (Figure 1).

### 2.2. The Cell-Type-Specific Genes of Ovarian Cells

Since scRNA-seq can detect gene expression in individual cells, it is possible to compare marker genes of clustered cell subpopulations with those of known cell types in the database to define cell types and discover new cell-type-specific genes [46]. Through analysis of scRNA-seq data, the marker genes of different types of ovarian cells in different species were initially discovered, and Table 1 shows the collective findings. The detailed information is provided in the following sections. Based on these cell-type-specific genes, it may be possible to isolate a certain cell population in vitro according to the needs of the research and to better study the function and characteristics of this cell population. In summary, these studies demonstrate the transcriptional specificity of ovarian cells.

## 3. A Single-Cell RNA Perspective of Ovarian Germ Cells

Ovarian germ cells mainly have two stages of development: fetal germ cells (FGCs) and oocytes [52]. Ovarian germ cells are essential for ovarian development and folliculogenesis [53]. If ovarian germ cells are lost, follicles cannot be formed, and the ovary loses its normal function and structure [54,55].In addition, some studies have reported that there is a finite oocyte pool for female fertility, which decreases with age [56,57]. However, there is a contradictory theory that rare oogonial stem cells (OSCs) exist in the ovarian cortex and can differentiate into oocyte-like cells in vitro [58,59]. Therefore, the existence of OSCs needs to be further explored. It is well-known that ovarian germ cells at various developmental stages are heterogeneous and have different transcriptomes, and scRNA-seq has the advantage of revealing transcriptomic profiles of individual cells to accurately classify cell types and reconstruct cell development by pseudotime analysis. Hence, scRNA-seq can facilitate the understanding of the transcriptome characteristics of individual ovarian germ cells, discover rare ovarian germ cell populations, explore changes in the expression profile during germ cell development, and overcome the limitations of histological classification. In this section, we present studies that have used scRNA-seq to characterize germ cells at distinct developmental stages.

### 3.1. A Single-Cell Perspective of Fetal Germ Cells

During ovarian development, fetal primordial germ cells (PGCs) originate from the yolk sac and migrate to the gonadal primordium early in fetal life. After undergoing several rounds of mitotic division and proliferation, oogonia are produced [39,60,61]. Then, the majority of oogonia enter meiosis to become oocytes, and dormant oocytes are arrested at the diplotene stage of meiosis I [62,63]. Fetal PGCs and oogonia can be collectively referred to as FGCs. scRNA-seq has been used for research on FGCs. Some studies have focused on the heterogeneity of FGCs and revealed the characteristics of FGCs in the process of the mitotic-to-meiotic transition and the initiation of meiosis. In this section, we focus on investigating the transcriptome of FGCs at a single-cell resolution using scRNA-seq.

#### 3.1.1. The Cell Subpopulations and Characteristics of Fetal Germ Cells

The healthy development of FGCs is a key prerequisite for the normal transmission of genetic information from parents to offspring [64,65]. There is great heterogeneity in the gene expression among individual FGCs [66,67], and the subpopulations and developmental trajectories of FGCs are not well-known. To identify clusters of FGCs, Zhao et al. [47] sequenced 19,363 single POU5F1-eGFP^+^ FGCs from E12.5, E14.5, and E16.5 female mouse gonads. *POU5F1* is a pluripotency transcription factor expressed in embryonic stem, early germ, and primordial germ cells [68,69]. After cluster analysis, researchers [47] found that FGCs can be divided into seven clusters according to the reported markers and differentially expressed genes. The subpopulations of FGCs included PGCs, oogonia, and preleptotene, leptotene, zygotene, early-pachytene, and late-pachytene cells. Each detection time point included several cell clusters, implying that FGCs developed asynchronously, which is consistent with another scRNA-seq study [48]. In addition, using magnetic- and fluorescence-activated cell sorting, c-KIT^+^ FGCs and larger c-KIT^−^ cells from 17 female embryos between 4 and 26 weeks after fertilization were isolated [39]. *C-KIT* has been reported to be a cell surface marker for germ cells [67]. Using the t-distributed random neighbor embedding method to analyze scRNA-seq data, Li et al. [39] discovered that FGCs can be divided into four developmental stages: mitotic FGCs, retinoid-acid (RA)-signaling-responsive FGCs, meiotic prophase FGCs, and oogenesis FGCs. Moreover, they found that *NANOG* might act as a specific marker gene for FGCs in the mitotic stage. In addition, RA-signaling-responsive FGCs specifically expressed *STRA8, ZGLP1, ANHX, ASB9*, and *THRA/BTR*. Furthermore, meiotic prophase FGCs are enriched in gametogenesis-specific genes, and *IL13RA2* may serve as a specific marker gene for meiotic prophase FGCs. As for oogenesis FGCs, they highly expressed not only genes related to the cytoskeletal organization, hormone-mediated signaling, microtubule-based movement, and oocyte development, but also programmed cell death and apoptosis genes, implying that some FGCs at that stage were undergoing apoptosis, which was consistent with previous observations that most germ cells became apoptotic as cysts broke down [70]. Oogenesis FGCs specifically expressed *PECAM1*, *ZP3*, and *OOSP2*.

To further observe the distribution of FGCs at different stages in the ovary, these researchers applied immunofluorescence staining with specific markers of FGCs at each stage to the 18-week embryonic ovary and found that mitotic and RA-signaling-responsive FGCs were preferentially located in the superficial cortex, while meiotic prophase and oogenesis FGCs were preferentially located in the deep layers of the ovary, suggesting spatial asynchrony in the FGCs. In addition, by utilizing the algorithm for the reconstruction of accurate cellular networks, they found that some important transcription factors that regulated the development of FGCs, such as *MERE* and *ZGLP1*, may regulate the development of RA-signaling-responsive FGCs, and *FIGLA* and *STAT1* may regulate the development of oogenesis FGCs [39].

Altogether, these studies reveal the unique gene expression characteristics of FGCs at different developmental stages, which will contribute to the isolation of FGCs, thus providing new avenues for understanding meiosis and oogenesis.

#### 3.1.2. Molecular Mechanisms of the Mitotic-to-Meiotic Transition and the Initiation of Meiosis in Fetal Germ Cells

FGCs migrate to the gonadal ridge during early embryonic development and enter meiosis after mitotic division [71]. The mitotic-to-meiotic transition and initiation of meiosis are critical steps in the successful development of gametes [72], but the molecular mechanisms of these two processes are not well understood. Based on scRNA-seq data, Zhao et al. [47] observed dramatic changes in the gene expression during the transition from mitosis to meiosis in FGCs, with the downregulation of pluripotency marker genes and a marked upregulation of oocyte marker genes. For example, they found that *REC8* may be involved in the transition from mitosis to meiosis and that the transcription factors *MSX1, MSX2, GATA2, CDX2, SOX4*, and *B-MYC* may be related to the initiation of meiosis. In addition, after analyzing scRNA-seq data, Li et al. [39] reported that RA synthesis may have important effects on female entry into meiosis. Furthermore, the scRNA-seq study of Zhao et al. [47] pointed out that the initiation of meiosis first occurs at E12.5. Interestingly, conducting detailed scRNA-seq analysis of more than 52,000 individual cells from the gonadal ridges and ovaries of E11.5–PD5 mouse embryos, Niu et al. [48] also demonstrated that the E12.5 FGCs contained a small number of meiotic cells, which is different from the previous view that female mouse germ cells begin meiosis between E13.5 and E14.5 [73], indicating that scRNA-seq had a unique advantage in finding rare cell populations. In addition, by performing scRNA-seq on 19,387 cells from E11.5–E14.5 mouse gonadal ridges and ovaries, Ge et al. [74] successfully depicted germ cell meiotic initiation using pseudotemporal ordering approaches.

These studies provide detailed information on the transition from mitosis to meiosis and gene expression at the initiation of meiosis in FGCs and offer new avenues for understanding the process of embryonic gonadal development. Defects in meiotic initiation fidelity may lead to some reproductive diseases [75,76]. Understanding the precise mechanisms of the mitotic-to-meiotic transition and initiation of meiosis of FGCs can help provide a theoretical basis for the future treatment of diseases related to abnormal gametogenesis.

### 3.2. A Single-Cell Perspective of Oocytes

Oocytes are the main determinants of embryonic developmental competence. Upon recruitment, oocytes undergo growth and maturation. [55,77,78,79]. Oocyte maturation can be affected by some ovarian-related diseases (for example, polycystic ovarian syndrome (PCOS)) [80]. In addition, some oocytes age with aging [81]. The oocyte undergoes dramatic changes in its transcriptional profile during these processes. In addition, with the increase in work and social pressures, the proportion of advanced pregnancies and infertility has been increasing over the past few decades [82,83]. The progress of assisted reproductive technology has brought hope of procreation to these populations [84,85]. However, the effects of in vitro culture techniques and age on the development and quality of oocytes should not be ignored [86,87]. Therefore, a new technological approach is needed to reveal potential mechanisms, thereby improving therapeutic effects. In addition, as the largest cell in the human body, the oocyte is rich in RNA, which gives it a unique advantage in scRNA-seq studies [88,89]. In this section, we review the latest progress in scRNA-seq in the field of oocyte developmental research.

#### 3.2.1. Oocyte Characteristics from Different Follicular Stages

As one of the most pivotal functional cell types in ovaries, oocytes play a decisive role in embryonic development, and their quality is extremely important for female fertility [90,91]. As follicles develop to a higher stage, oocytes constantly change [92]. The emergence of scRNA-seq makes it possible to study the gene expression of oocytes at the single-cell level, which enriches the understanding of oocyte development. Some scRNA-seq studies demonstrated that human oocytes expressed *GDF9, ZP3, DDX4, SYCP3, DAZL, FIGLA, OOSP2, ZP2, SOX30*, and *ZAR1* [34,49,50]. Furthermore, cynomolgus monkey oocytes specifically expressed *GDF9, ZP3, DDX4, SYCP3, LMOD3, RBM46*, and *NET01* [45], while mouse oocytes highly expressed *DDX4* and *DAZL* [36,37,38]. In addition, by integrating single-cell sequencing data from cells in the adult ovarian cortex with diameters less than 35 µm and scRNA-seq data from the fetal ovary, Wagner et al. [34] found that compared with fetal PGCs, the pluripotency marker genes *NANOG* and *POU5F1* were less expressed in most developmental stages of adult oocytes, which may be associated with the stemness of PGCs [49,66].

Additionally, Zhang et al. [49] analyzed the scRNA-seq data of human oocytes and found that the marker genes of oocytes from various follicular stages exhibited different expression patterns. For instance, *DDX4, ZP2, ZP3*, and *ZP4* showed stable expression in oocytes from follicles at different stages of development, but *ZP1, GDF9*, and *H1FOO* were gradually upregulated with oocyte maturation. Data from single-cell transcriptome sequencing of monkey oocytes also showed that the expression of *ZP1* and *GDF9* was gradually upregulated during follicular development [45]. Furthermore, this study showed that the expression of most meiosis-related genes in the oocyte was upregulated as the oocyte developed, and these genes were mostly expressed in oocytes from the antral and preovulatory follicles, which was consistent with a previous study showing that arrested oocytes gradually acquired the capacity to resume meiosis during the transition from the preantral to antral stage [93]. Moreover, with the maturation of oocytes, the activity of some ten-eleven translocation family genes was low in adult oocytes at all follicular stages, and *DNMT1, DNMT3A*, and *DNMT3B* expression increased gradually [49]. The expression of *DNMT1* and *DNMT3A* in monkeys gradually increased with the development of oocytes [45]. Considering that ten-eleven translocation family proteins are associated with active DNA demethylation, and the DNA methyltransferase family is related to the maintenance of DNA methylation [94,95], the findings indicated that the oocyte maturation was accompanied by an increased DNA methylation.

More importantly, by analyzing the regulatory network of transcription factors in human oocytes at each follicular stage, Zhang et al. [49] found that *SOX30* may play an important role in the transition from primordial to primary follicles, and *SOX13* and *SOX15* may be necessary for antral follicle formation. Wang et al. [45] elaborated on the detailed stage-specific regulatory networks of transcription factors that regulate cell-type-specific markers of oocytes from various follicular stages. *ELF4* and *FOS* may be essential for oocytes from primordial follicles. In addition, *RPS4X* and *FIGLA* may play vital roles in oocytes from primary follicles. Moreover, by comparing the single-cell transcriptome profiles of oocytes to the previously reported RNA data of mice, researchers found that humans not only shared some genes with mice, but also had some special regulatory mechanisms in oogenesis, which may provide a better understanding of oocyte development between humans and mice [49].

Understanding the genetic characteristics during oocyte development will provide a basis for elucidating the molecular mechanisms of folliculogenesis. It is noteworthy that the oocytes were collected by manual micromanipulation in the studies by Zhang et al. and Wang et al. [45,49]. The reasons for this may be as follows: (1) Oocytes are too fragile, and a limited number are available; (2) Although the large volume of the oocyte results in a relatively rich RNA content that provides a good basis for single-cell sequencing, the large oocyte diameter also makes it difficult for the cell to pass through narrow nozzles without damage. For example, fluorescence-activated cell sorting, which is a plate-based platform, captures cells less than 50 μm in diameter with minimal damage, whereas droplet-based platforms capture cells less than 30 μm in diameter [96,97]. The diameter of oocytes from primordial follicles is approximately 35 µm [98]. After several months of growth, the diameter can reach 120 µm [99]. Therefore, the techniques are not suitable for sequencing all oocytes. Consequently, the size of the nozzle needs to be further improved for future single-cell detection.

#### 3.2.2. The Characteristics of In Vitro Matured Oocytes

Oocyte maturation is a determinant step for embryo development, providing sufficient energy and nutritional materials for early-stage embryonic growth, and is influenced by many factors, among which the environment is of great importance [100,101]. In vitro matured (IVM) oocytes have poor developmental competence, and the unsynchronized cytoplasmic maturation and nuclear saturation may be the reason. However, the underlying molecular mechanisms remain ambiguous [102]. Using scRNA-seq, Zhao et al. [103] studied three IVM oocytes and three in vivo (IVO) matured oocytes. They showed that exposure to the in vitro environment can lead to a decline in the activity of CoA-related enzymes, such as *ACAT1* and *HADHA*, ultimately contributing to a decrease in energy metabolism. The energy metabolism capacity is known to play a significant role in regulating the gene expression, protein translation, and protein modification, and a decrease in the oocyte energy metabolism capacity may impair the developmental potential [101,104]. These findings imply that targeting genes involved in energy metabolism may help improve the adverse outcomes of IVM oocytes. Furthermore, the increased level of NADP+ resulting from the high expression of nicotinamide nucleotide transhydrogenase enhances the ability of humans to repair DNA double-strand breaks, thus maintaining euploidy. This may be a key reason why, although the developmental potential of some fertilized eggs is low, they can complete the normal maturation process in vitro. The results will likely yield further insights into the mechanisms of human oocyte maturation, thus providing benefits to some patients with infertility.

#### 3.2.3. The Characteristics of Oocytes from Aged Females

It is well-known that the quality of oocytes decreases with age, thus impairing the embryo’s developmental potential [57]; however, the molecular mechanisms have not been explained clearly.

Recently, using the scRNA-seq technique, Zhang et al. [105] discovered that compared to in oocytes from younger women, genes related to oxidative stress, transcriptional activation, and immune function in those from older women were upregulated. Additionally, a single-cell study sequenced a total of six matured oocytes from younger (≤30 years) and older (≥40 years) patients and found that some genes associated with oxidative stress were dramatically upregulated in oocytes from older women compared to in those from younger women, while the gene *TOP2B*, devoted to promoting double-strand break repair after oxidative stress, was downregulated in oocytes from older women [105]. In addition, Wang et al. [45] analyzed high-quality transcriptomes of 418 oocytes collected from four young and four aged cynomolgus monkeys at a single-cell resolution. They demonstrated that antioxidant genes, such as *GPX1* and *GSR*, were downregulated in the oocytes of aged cynomolgus monkeys, which could be responsible for the increased oxidative damage during ovarian aging.

Considering that oxidative stress may damage the oocyte proteome with negative consequences for meiosis, fertilization, and embryonic development [106,107], these results provide a new way to evaluate the quality of oocytes in older women and elucidate the causes of ovarian aging at the molecular level, thus contributing to the discovery of new biomarkers to improve the oocyte quality. As more women worldwide are delaying the age of reproduction, resolving the mechanism of decreased oocyte quality with increasing age will be helpful for family health and social stability.

#### 3.2.4. The Characteristics of Oocytes from Polycystic Ovarian Syndrome Patients

The oocytes in PCOS patients are often of poor quality, leading to lower fertilization, cleavage, and implantation rates [108]. It is essential to elucidate the mechanism behind this in detail. By analyzing the scRNA-seq data from 14 oocytes from seven healthy fertile women and 20 oocytes from nine patients with PCOS at the germinal vesicle (GV) stage, metaphase I stage, and metaphase II (MII) stage, Qi et al. discovered that some genes associated with mitochondrial function (for example, oxidative phosphorylation), such as *COX6B1*, *COX8A*, *COX4l1*, and *NDUFB9* were prematurely activated at the GV stage of PCOS oocytes, whereas it occurs at the MII stage in healthy oocytes [109]. Mitochondria plays an essential role in the oocyte maturation, meiotic spindle assembly, fertilization, and subsequent preimplantation embryogenesis [110]. Abnormal function of the mitochondria may account for the low-quality oocytes in PCOS patients, which provides a new idea for improving reproductive outcome of PCOS patients.

### 3.3. A Single-Cell Perspective of Oogonial Stem Cells

A heated debate about whether there are OSCs in the female ovary that can produce new oocytes has been consistently ongoing. Some studies have suggested that mammalian oocytes are formed in the early stages of fetal life, and their finite number represents female fertility, which decreases with increasing female age [111,112]. Nevertheless, some researchers have recently indicated that OSCs found in the ovarian cortex of female mammals have the ability to self-renew, clone, expand, and differentiate into oocytes [113,114]. When these OSC-derived oocytes combine with sperm, offspring can be produced, which might provide a basis for the clinical application of OSCs [115,116,117]. However, little is known regarding the biological characteristics of OSCs. Some studies have reported that DEAD box polypeptide 4 (DDX4) can be used to sort OSCs [118,119]. Recently, using immunomagnetic bead sorting with DDX4-based antibodies to enrich OSCs, Wu et al. [117] identified the developmental characteristics of transplanted OSCs by single-follicle RNA sequencing of follicles transplanted with GFP-positive OSCs in recipient and wild-type (WT) mice. The results showed that OSCs expressed *DDX4, DPPA3, IFITM3, POU5F1, DAZL*, and *PRDM1*. Single-follicle RNA data indicated that the expression profiles of preantral follicles of WT mice were similar to those of OSC-derived preantral follicles. In addition, small antral follicles of WT mice and OSC-derived antral follicles showed similar gene expressions, indicating that follicles from WT mice and OSC-derived follicles have analogous developmental mechanisms. For instance, they found that the PI3K-AKT signaling pathway plays an essential role in the development of OSC-derived preantral follicles to small antral follicles, which is consistent with previous studies on the follicular development [120,121].

Currently, scRNA-seq has emerged as a valuable tool for identifying rare cell populations [25]. To explore whether OSCs are in the ovary, Wagner et al. utilized scRNA-seq to investigate the transcriptome of more than 24,000 cells in the adult ovarian cortex at a single-cell resolution. Using unbiased cluster analysis, they discovered that most cells collected by the DDX4 antibody belonged to perivascular cells that did not express DDX4 transcripts, rather than to OSCs. Furthermore, OSCs were not found in the ovarian cortex [34]. Given that single-cell transcriptome sequencing technology has significant advantages in discovering rare cell types, the process of obtaining single cells may have harmful effects on cellular activity and may even lead to the death of rare cells [122]. In addition, Bhartiya et al. [123] thought that the size of OSCs was so small that they needed to be collected with high-speed centrifugation, such as at 1000 g, and a low speed does not allow OSCs to settle. Hence, it is possible this is the reason why the current study did not capture OSCs. In the future, improving the process of collecting single cells and expanding the sample size may be an effective way to confirm the presence of OSCs in the ovary.

## 4. A Single-Cell RNA Perspective of Ovarian Somatic Cells

Ovarian somatic cells comprise several cell types, including granulosa cells (developed from pregranulosa cells), stromal cells, endothelial cells, vascular smooth muscle cells, and immune cells. In this section, we describe the latest scRNA-seq findings for ovarian somatic cells.

### 4.1. A Single-Cell Perspective of Pregranulosa Cells

Pregranulosa cells (pre-GCs) originate from ovarian surface epithelium cells and are precursors of granulosa cells [124,125]. It is believed that pre-GCs are essential for the development of primordial follicles and the establishment of ovarian reserves [126,127]. By sequencing 563 XX *NR5A1-GFP*^+^ somatic cells from E10.5, E11.5, E12.5, E13.5, E16.5, and PD6 mouse ovaries at the cellular level, Ste’vant et al. [128] found that pre-GCs were first observed as early progenitors on E11.5, using transcriptome analysis. Pre-GCs remained in an early differentiation stage, expressing stem-cell-related genes until E16.5, and then continued to differentiate through folliculogenesis. Furthermore, they revealed that pre-GCs expressed genes associated with lipid metabolic processes and hormone secretion, implying that pre-GCs had the potential to provide hormones at the fetal stage, which was in accordance with the results of a previous study [129].

In addition, two types of pre-GCs have been reported in mammalian embryonic ovaries [130,131]. To explore the cellular origin, division timing, and gene expression profiles of pre-GCs, Niu et al. [48] obtained single-cell transcriptomic data of over 52,500 single cells from between E11.5 and postembryonic day 5 from 10–18 mouse gonads/ovaries. They demonstrated that pre-GCs specifically express *WNT4, WNT6, KITL*, and *FOXL2*. They also revealed that bipotential pre-GCs (BPG) originated from bipotential precursors, differentiated into wave 1 follicles that began to develop immediately after birth in the medullar region and were related to the onset of fertility, while epithelial pre-GCs (EPG) derived from the ovarian epithelial progenitor cells became the granulosa cells of the primordial follicles in the ovarian cortex, which represented the ovarian reserve. Bipotential precursors were derived from the ovarian epithelial progenitor cells (Figure 2). The two cell types had similar, but distinct, gene expression profiles. For instance, bipotential precursors had a nonmitotic signature, with low MKI67 and *HIST1H1AP* but high *CDKN1B* expressions, and the ovarian surface epithelium had mitotic characteristics, with high *MKI67* expression. Furthermore, BPG cells expressed *FOXL2* early, while EPG cells expressed *LGR5* and abundant *FOXL2* until after birth, which was consistent with the results of the single-cell transcriptome sequencing in the study by Wang et al. [36]. Moreover, using lineage tracing, they found that with the development of the embryo, cortical BPG cells were replaced by EPG cells, and EPG cells never significantly resided in the ovarian medulla.

Considering that the recent emergence of spatial transcriptomics as a method that can be used to depict gene expression profiling with spatial localization information in tissues [132,133], future single-cell profiling with spatially resolved transcriptomes will provide critical insights into pre-GC development. Additionally, these studies have allowed insight into the evolution of granulosa cells (GCs), so as to provide important resources for the further study of the molecular mechanisms of follicular development.

### 4.2. A Single-Cell Perspective of Granulosa Cells

GCs play a critical role in maintaining the oocyte quality and endocrine ovarian function [134]. During oocyte development and folliculogenesis, GCs undergo proliferation and differentiation [135]. Importantly, GCs also participate in ovulation [136,137]. Moreover, many GCs become atretic during follicular development [138]. The quality of GCs declines with aging [139]. Recently, scRNA-seq was conducted to characterize GCs at different cellular stages. This section provides a short overview using scRNA-seq to shed light on transcriptome changes related to GC states.

#### 4.2.1. The Characteristics of Granulosa Cells from Different Stages of Follicles

As a major cell type of ovarian somatic cell, GCs not only provide nutritional and mechanical support for oocytes through gap junctions, but also regulate oocytes by paracrine signals, which play a vital role in folliculogenesis and oocyte development [140,141]. GCs are closely associated with ovarian function and fertility. Exploring the molecular characteristics of GCs can further improve our understanding of follicular development.

When it comes to the possible marker genes of GCs, some scRNA-seq studies suggested that human GCs expressed *AMH, WT1, FOXL2, STAR, SERPINE2, GSTA1, CYP11A1, INHBA, CDH2, GJA1*, and *TNNI3* [34,35,39,49,50], nonhuman primate GCs expressed *AMH, WT1*, and *INH4* [45], and mouse GCs expressed *AMH, AMHR2, KITL, FSHR*, and *CYP19A1* [36,38,51]. Moreover, Zhang et al. [49] observed that genes related to steroid production are dynamically expressed in GCs during folliculogenesis. For example, the expression levels of *HSD17B1, HSD3B2*, and *NR5A1* in GCs gradually increase with follicle growth, indicating that steroid hormones are important for follicular development. Furthermore, their analysis revealed that transcription factor regulatory networks in GCs may be involved in the regulation of folliculogenesis. For instance, *GREB1, NFKB1, MEF2A, PIASI, FOSL2, KLF13*, and *PRDM4* may potentially participate in regulating primordial activation, which will provide new clues for further studies on folliculogenesis.

As for GC subpopulations, Zhang et al. [49] performed scRNA-seq on adult GCs from primordial to preovulatory follicles and then analyzed the data using the principal component unsupervised method. They showed that GCs could be divided into five clusters according to the stages of follicular development (i.e., primordial, primary, secondary, antral, and preovulatory follicular GCs) and presented a subset of stage-specific genes as candidate cell-type-specific markers for GCs during folliculogenesis, including *MGP, PTK6, KIF20A*, and *VTN*, that might separately serve as signature genes for primary, secondary, antral, and preovulatory follicular GCs. However, this was demonstrated using sequencing data based on the gene expression from a small number of GCs from each follicular stage, and the reliability requires validation through further experiments. They also showed that the aggregation of GCs at different stages of follicular development showed some overlap, with more overlap between primary and secondary follicular GCs, indicating that the transcriptional patterns of the two stages were similar. GCs of primordial follicles are distinct from those of antral follicles. Similarly, Wagner et al. [34] also discovered there was no overlap between primordial and antral follicular GCs from the inner ovarian cortex, implying that the more separated the GC developmental stages, the more different the gene expression profiles.

In addition, scRNA-seq studies demonstrated that mural and cumulus GCs from small antral follicles gathered together showed high expression levels of *WT1* and *EGR4* but low expression levels of *VCAN* and *FST.* However, when mural and cumulus GCs from selective follicles were divided into two groups, human cumulus GCs expressed high levels of *IGFBP2, INHBB, IHH, VCAN, FST*, and *HTRA1*, while human mural GCs showed high expression levels of *CYP19A1, KRT18, AKIRIN1, LIHP*, and *CITED2* [35,50]. In mice, cumulus GCs from selective follicles can be identified by *HAS2* and *NPR2*, whereas mural GCs from selective follicles can be identified by *FSHR, BMPR2*, and *NPPC* [51]. It has been reported that early GCs undergo metabolic reprogramming as they become cumulus GCs [51], suggesting that mural GCs and cumulus cells from early antral follicles undergo differentiation.

#### 4.2.2. The Characteristics of Granulosa Cells during Ovulation

The ovary experiences repeated ovulation during the reproductive period, and ovulation is driven by the surge of luteinizing hormone (LH) [142]. An LH surge can result in an altered cell function and gene expression in mural and cumulus GCs. When stimulated by ovulatory signals, luteinizing GCs undergo a transition from proliferation to differentiation and expansion [143], and GCs in preovulatory follicles will express ovulation-related genes, such as *PGR* and *PTGS2*. Additionally, previous studies have shown that GCs play an important regulatory role in ovulation [136,144], the role of individual GCs or certain GC subsets in ovulation has not been revealed, and the related mechanisms need to be further discussed. The advent of scRNA-seq has made it possible to study individual cells or choose a subpopulation for analysis based on the gene expression.

To investigate the characteristics of GCs during ovulation, Dong et al. [91] performed scRNA-seq analysis of hundreds of cumulus cells from two women with normal ovarian function and found a CD24(+) cumulus GC subpopulation, which played a crucial role in triggering ovulation. The CD24(+) cumulus GC subpopulation can activate the EGFR-ERK1/2 pathway and induce the expression of prostaglandin synthases and transporters, such as *PTGS2, PLA2G4A, SLCO2A1*, and *ABCC4*. PGR has been reported to have an important role in ovulation [145,146]. Additionally, repeated ovulation is accompanied by inflammation, and the ovary can maintain a dynamic balance for several years [147,148]. To explore the mechanism by which the ovary protects itself from repeated ovulatory inflammation, Park et al. [149] applied scRNA-seq to the ovaries of *ESR2-PGR* knockout (KO) and WT mice. Their results demonstrated that *PGR* can indirectly inhibit the synthesis of *PTGS2* and *PGE2* in GCs by inhibiting *NF-κB*, thereby reducing the ovarian damage caused by oxidative stress and the DNA damage induced by ovulatory inflammation.

Given that GCs are involved in ovulation and that anovulation is an important cause of infertility [84], a better understanding of the characteristics of GCs during ovulation may be helpful for pregnancy outcomes in the field of reproductive medicine.

#### 4.2.3. The Characteristics of Granulosa Cells in Atretic Follicles

In female mammals, more than 99% of follicles undergo atresia at various stages [150]. GCs play a crucial role in regulating follicular atresia [151]. Using scRNA-seq, FAN et al. [50] found that GCs from atretic follicles highly express genes related to myeloid leukocyte activation, such as *FCER1G, CD53, AIF1*, and *CX3Cl1*. Macrophages have been observed in GC layers from atretic follicles, and excessive T-helper 1 in the ovary leads to follicular atresia, implying that immune cells play a role in follicular atresia [152,153,154,155]. In addition, another scRNA-seq study by FAN et al. [35] discovered that GCs in the early stages of atresia had lower levels of *GJA1* and *CDH2*. *GJA1* and *CDH2* are associated with cell connections [156,157,158], indicating that the cell–cell communication in atretic follicles decreased. Follicular atresia is important for maintaining homeostasis of the ovarian internal environment, and a better exploration of the characteristics of atretic GCs will facilitate the understanding of the molecular mechanisms of follicular atresia.

#### 4.2.4. The Characteristics of Granulosa Cells from Aged Females

The characteristics of GCs change with the progression of ovarian aging. Some studies have shown that oxidative stress causes damage to GCs during ovarian aging, thus affecting the normal development of oocytes and influencing the fertility of women [139,159]. The analysis of scRNA-seq results of juvenile and old monkey GCs showed that the expression of apoptosis-related genes in GCs from aged female ovaries was upregulated, while the expression of genes related to oxidoreductase activity was downregulated. In addition, the transcriptional regulatory network demonstrated that key genes, including *ELF4* and *FOSB*, which are crucial in regulating reductase-activity-related genes such as *IDH1* and *NDUFB10*, were downregulated [45]. A decrease in antioxidant enzyme activity leads to a damaged antioxidant response, resulting in the enhanced production of reactive oxygen species in GCs [160,161]. Therefore, the antioxidant capacity of GCs is impaired with increasing age, contributing to a decline in the GC quality. The scRNA-seq study more specifically explains the damage mechanism of oxidative stress in senescent GCs, which will help put forward anti-aging strategies in the future.

### 4.3. A Single-Cell Perspective of Ovarian Stromal Cells

Approximately 83% of ovarian cortex cells are classified as stroma [34], and the majority of ovarian stroma are stromal cells [162]. Furthermore, “ovarian stromal cells” comprise multiple cell populations [163]. The emergence of scRNA-seq makes it possible to better characterize stromal cells. Using this method, researchers found that some genes, such as *DCN, COLLA1, LUM, APOE*, and *COL3A1*, could serve as cellular markers for human stromal and theca cells [35,50]. Another scRNA-seq study discovered that human stromal cells expressed mesodermal lineage markers (*DCN* and *PDGFRA*) and extracellular matrix proteins (*COLLA1* and *COL6A1*) [34]. In nonhuman primate ovarian stromal cells, cellular markers have been reported, such as *TCF21* and *COLLA2*, and some cells of this population show high expression levels of the theca cell marker steroidogenic acute regulatory protein [45]. Because theca cells are generated from stromal cells [164], they may have the same cell markers. In addition, mouse stromal cells specifically express *TCF21* and *NR2F2* [36]. These studies demonstrated the characteristics of stromal cells under normal physiological conditions.

Additionally, stromal cells from atretic follicles show high levels of *XBP1* and *SELK*, which participate in endoplasmic-stress-induced apoptosis [35]. Apoptosis is the underlying mechanism of follicular atresia [165], implying that the apoptosis of stromal cells may play a role in follicular atresia. More importantly, stromal cells from atretic follicles express genes related to the complement system, such as *C1R, C1S*, and *C7*. As the complement system plays an important role in immune and inflammatory responses [166], activation of the complement system may contribute to ovarian remodeling and the maintenance of ovarian homeostasis. Given that many follicles undergo atresia before reaching ovulation in every ovarian cycle, an understanding of the features of stromal cells from atretic follicles can provide new insights into the process of ovarian and follicular remodeling.

Furthermore, stromal cells obtained from primary ovarian tumors were enriched for upregulated extracellular matrix genes and genes associated with the epithelial-to-mesenchymal transition (EMT) [167]. The EMT plays an important role in tumor cell metastasis. This characteristic is associated with poor ovarian cancer prognosis [168].

scRNA-seq makes it more convenient to understand the features of stromal cells; however, these findings have some limitations. For example, stromal cells have only been collected from the inner ovarian cortex in most studies. Collecting ovarian tissue on a larger scale and at a deeper anatomical level in the future will be helpful for characterizing stromal cells and for a more comprehensive understanding of ovarian stromal cells.

### 4.4. A Single-Cell Perspective of Ovarian Smooth Muscle Cells

Smooth muscle cells (SMCs) were discovered in the ovary around the follicles, corpora lutea, atretic follicles, and between groups of interstitial cells [44]. They play an essential role in ovulation, regulating the growth and development of the corpus luteum and the collapse of the ruptured follicle [169,170,171]. scRNA-seq technology has contributed to identifying markers. Two studies on SMCs from the inner cortex of adult human ovaries found that they can be recognized by *TAGLN* and *RGS5* [35,50]. Another scRNA-seq study of the human ovarian cortex identified SMCs expressing *TAGLN, RGS5, MYH11, MCAM*, and *RERGL* [34]. Single-cell transcriptomic analysis of nonhuman primate ovaries revealed that SMCs specifically expressed *DES* and *ACTA2* [45]. These studies have contributed significantly to the mapping of ovarian SMC signatures.

### 4.5. A Single-Cell Perspective of Ovarian Endothelial Cells

Ovarian endothelial cells (OECs) play an essential role in neovascularization, follicular development, and the formation and function of the corpus luteum [9,172,173]. scRNA-seq has enabled significant progress in the understanding of OECs. Several scRNA-seq studies have demonstrated that human OECs can be identified based on the expression of *CDH5 (CD144), vWF, CLDN5, PECAM1 (CD31)*, and *CD34* [34,35,39,50]. An investigation of nonhuman primate ovaries also confirmed that OECs can be identified by the high expression of *CDH5* and *vWF* [45]. In addition, three single-cell transcriptomic studies of mouse E16.5 to PD3 ovaries revealed that OECs can express *APLNR* and *EGFL7* [36,37,38]. Understanding the heterogeneity and clarifying the pathological contributions of OECs remain important tasks.

### 4.6. A Single-Cell Perspective of Ovarian Immune Cells

Ovarian immune cells comprise macrophages, monocytes, B lymphocytes, T lymphocytes, and natural killer cells [163]. Immune cells play a key role in supporting optimal ovarian function and contribute to folliculogenesis, ovulation, and corpus luteum formation and regression [174,175]. Recently, scRNA-seq has yielded significant achievements in the characterization of ovarian immune cells. Two scRNA-seq studies of humans showed that *CD53, CXCR4, CD69*, and *ITGB2* may be marker genes of ovarian immune cells; human innate immune cells highly express *CD68* and *IFI30*; human ovarian T lymphocytes specifically express *CD2, CD3G*, and *CD8A*; and human antigen-presenting cells specifically express *CD14, HLA-DRA, B2M*, and *HLA-DQB1* [34,35]. A study of nonhuman primates suggested that natural killer cells expressed *CD3D* and *KLRB1*, whereas macrophages expressed *CD68* and *CD14* [45]. In mice, ovarian immune cells specifically express *ELANE, MPO*, and *TYROBP* [36,37,38]. A scRNA-seq investigation of mouse follicles indicated that T lymphocytes can be marked by *AW112010* and *CD3G*, B lymphocytes can be marked by *IGHM* and *CD37*, and monocytes or monocyte-derived cells can be marked by *CD14* [51]. These studies provided an experimental basis for exploring ovarian immune cells in vitro.

## 5. Applications of Single-Cell Transcriptome Sequencing in Ovarian Intercellular Interactions

Serving as a heterogeneous multicellular tissue, the ovary has complicated and coordinated intercellular communications [176]. Researchers have long realized the importance of the interaction between germ and ovarian somatic cells in maintaining a regular ovary and promoting oogenesis. Interactions between FGCs and fetal somatic cells are critical for ovarian development. Bidirectional communication between oocytes and GCs is essential for primordial follicular formation and folliculogenesis. The growth and differentiation of oocytes rely on the support provided by GCs, which can regulate a wide variety of oocyte functions, including metabolism, meiotic arrest and resumption, and cytoskeletal rearrangements [177]. Conversely, the oocyte can regulate the proliferation and differentiation of GCs [178,179]. GCs and oocytes can communicate with each other through different signaling modalities, such as direct contact, ligand–receptor interactions, paracrine signaling pathways, gap junctions, and other junctional contacts via transzonal projections, gap junctions, and receptor tyrosine kinases [180]. Some canonical pathways have been reported to be involved in communication between germ and somatic cells, such as the TGF-β, KIT, and NOTCH pathways [181,182,183,184,185].

scRNA-seq is reported to have a great advantage in analyzing the intercellular interactions of complex organs, such as the liver, lung, and brain [186,187,188,189]. It can identify communication networks between different cells. The ovary is an essential organ with complex cell–cell interactions, and it is necessary to identify the mechanisms that participate in ovarian cellular communication. In what follows, we describe insights gained from recent scRNA-seq that cast a novel light on the molecular characteristics of ovarian intercellular crosstalk.

### 5.1. Landscape of Interactions between Female Human Embryonic Germ Cells and Gonadal Somatic Cells

FGCs can develop into functional oocytes only when they cooperate well with fetal ovarian somatic cells. In addition, when germ cells are lost after the onset of meiosis, female supporting cells transdifferentiate into Sertoli-like cells [53,190]. The interaction between germ and somatic cells is important for the maintenance of the ovarian structure and function. Germ and somatic cells communicate with each other mainly via ligand–receptor signaling [185,191]. By analyzing the dynamic gene expression in key cell signaling pathways involved in the communication between FGCs and fetal ovarian somatic cells in scRNA-seq data, including the NOTCH, TGF-β, and KIT signaling pathways, Li et al. [39] demonstrated that FGCs expressed the ligands *DLL3* and *JAG1* for the NOTCH pathway, which bind to the NOTCH2 receptor expressed by gonadal somatic cells, then activated the target gene *HES1* on gonadal somatic cells. For the TGF-β pathway, the ligand *BMP2*, expressed by mural and late GCs, activates the TGF-β pathway by acting on the target genes *ID2* and *ID3*, which are expressed by RA-signaling-responsive, meiotic prophase, and oogenesis FGCs. For the KIT signaling pathway, *KITL*, expressed by gonadal somatic cells, initiated the KIT pathway by acting on the receptor expressed by embryonic germ cells (Figure 3A). Elucidation of the molecular characteristics of key cell signaling pathways in the interaction between embryonic germ and gonadal niche cells will help to better interpret the process of embryonic ovarian development.

### 5.2. Landscape of Crosstalk between Oocytes and Granulosa Cells during Primordial Follicular Assembly

The assembly of primordial follicles (PFs) occurs when oocytes are encapsulated by squamous GCs [192]. The formation of PFs is the basis of female fertility [193]. PF assembly occurs during early postnatal development in rodents [194]. This coordinated process relies on the communication between oocytes and GCs [195].

To further dissect the characteristics of cellular interactions in PF assembly, Wang et al. [36] analyzed scRNA-seq data of mouse ovaries from postnatal PD0 and PD3. Their data characterized canonical cell-to-cell signaling pathways between oocytes and GCs involved in PF formation, such as the NOTCH, TGF-β, and KIT signaling pathways. The NOTCH signaling ligands *DLL3, JAG1*, and *JAG2* are mainly expressed by oocytes, whereas the *NOTCH2* receptor is expressed by GCs. The target genes *HES1* and *RBPJ* were expressed in both oocytes and GCs, although the expression of *HES1* and *RBPJ* in oocytes was transient. For the TGF-β signaling pathway, its important component *GDF9* exists in oocytes, while its receptors *BMPR1A* and *BMPR2,* as well as its target genes *ID1, ID2,* and *ID3*, were found both in oocytes and GCs, implying that the PF formation is regulated by autocrine and paracrine effects. Recently, another scRNA-seq study described the destruction of the TGF-β signal between oocytes and GCs in postnatal pups following maternal 2-ethyhexyl phthalate exposure, and PF formation was impaired [37], which further reflects the important role of the TGF-β pathway and cellular communication in the participation of the PF assembly. As for KIT signaling, GCs express its ligand, *KITL*, and oocytes express its receptor, *KIT* [36]. Except for those classic pathways, gap junctions also participate in the interactions between oocytes and granulosa cells during the formation of PFs, with oocytes expressing *GJC1* and *GCs* expressing *GJA1* [36,196] (Figure 3B). In addition, Wang et al. [36] discovered other oocyte and GC pathways involved in the PF assembly, such as FoxO and Hippo signaling. The importance of Hippo signaling in the PF assembly has also been reported in another study [38]. By performing scRNA-seq on PD0 and PD3 ovarian tissues from mice whose mothers were administered daily zearalenone from 16.5 days postcoitum, researchers found that their Hippo signaling was affected, which led to the blockage of the PF assembly [38]. Furthermore, Hippo signaling has been reported to crosstalk with NOTCH and TGF-β signaling. The Notch intracellular domain can enhance the activity of *YAP/TAZ* (the effector of Hippo signaling), and *SMAD* (a component of TGF-β signaling) can bind to *TAZ* to activate Hippo signaling [197]. This indicates that multiple pathways can cooperate to regulate the PF assembly. These studies contribute to a deeper understanding of the mechanisms of cellular interactions in the PF formation.

### 5.3. Landscape of Interactions between Oocytes and Granulosa Cells during Folliculogenesis

Folliculogenesis refers to follicles growing from the primordial to the antral stage [198]. The dormancy and activation of primordial follicles are critical steps in folliculogenesis, which is important for the maintenance of the female reproductive ability. Most primordial follicles remain dormant, and only a few primordial follicles are activated to become growing follicles [199]. Women may progress to premature ovarian insufficiency when primordial follicles are massively overactivated [200,201]. The activation of primordial follicles does not involve the regulation of gonadotropins and may depend on the transduction of endogenous signals [92,202,203]. To further explore the potential molecular mechanisms involved in primordial follicular activation, Zhang et al. [49] compared scRNA-seq data of pairwise oocytes and granulosa cells from primordial and primary follicles. They found that the insulin, gonadotropin-releasing hormone, neurotrophin, and PI3K-mTOR signaling pathways may participate in coordinating the crosstalk between oocytes and GCs, which play an important role in the transition from the primordial to the primary follicular stage. It is worth noting that previous studies also reported that the insulin, PI3K-mTOR, and neurotrophin signaling pathways were involved in the recruitment of primordial follicles [183,204,205,206,207]. Because primordial follicles serve as a woman’s ovarian reserve, these results indicate that the further study of the gene expression signatures of critical signaling pathways in the activation of primordial follicles will help future research on the regulation of female fertility.

After primordial follicles are recruited into the growing follicular population, a few primary follicles sequentially become secondary follicles, antral follicles, preovulatory follicles, and eventually experience ovulation [208,209]. Follicular development depends on signaling interactions between oocytes and GCs. To study the interactions between oocytes and granulosa cells during folliculogenesis, Zhang et al. [49] focused on the expression of key ligands, receptors, and target genes in several important signaling pathways closely related to folliculogenesis, and the results showed that the NOTCH pathway is initiated by the key ligands *DLL3* and *JAG2* expressed by oocytes. *NOTCH2* and *NOTCH3*, as well as their target gene *HES1*, are expressed by GCs. Oocytes can regulate GC proliferation and differentiation via NOTCH signaling. As for the TGF-β signaling pathway, the important components, *GDF9* and *BMP15*, are expressed by oocytes. However, the receptor *BMPR2* and downstream target gene *ID3* were expressed in both oocytes and GCs. The TGF-β signaling pathway is activated by autocrine and paracrine mechanisms to regulate folliculogenesis. In addition, the key KIT pathway is activated by *KITLG* expressed by GCs via paracrine effects. Apart from these ligand–receptor interactions, gap junction intercellular communication also plays an essential role in ovarian folliculogenesis [210]. Zhang et al. [49] discovered that oocytes expressed *GJA3* and *GJC1*, whereas GCs expressed *GJA1* and *GJA5* (Figure 3C).

Clarification of the expression of important molecules in key signaling pathways will be beneficial to further deepen our understanding of folliculogenesis.

## 6. Conclusions and Future Perspectives

After several years of development, scRNA-seq technology has become a powerful tool for deconstructing complex organizations. The advent of single-cell transcriptomics has made it possible to study ovarian cell development and interactions at single-cell resolution. ScRNA-seq has greatly contributed to the classification of ovarian cell subpopulations, resolving the high heterogeneity among ovarian cells, exploring cell developmental trajectories, and uncovering cell-to-cell interactions in the ovary [36,50,51,74], undoubtedly promoting a new understanding of ovarian physiology at the molecular level.

However, there are some limitations to scRNA-seq. First, aspects of the single-cell isolation process, such as the enzymatic treatment, digestion, mechanical stress, and freezing and thawing of cells, may alter the expression of some genes [19,211,212,213,214,215]. This may lead to the high expression of dissociation-related genes in some ovarian cells [35]. Hence, more attention should be paid to improving the fidelity of single-cell sequencing. Second, scRNA-seq technology can rarely detect noncoding RNA, and noncoding RNA is vital for regulating ovarian development [25,216,217]. The coverage of scRNA-seq also needs to be urgently improved. Finally, scRNA-seq can identify subpopulations but loses information on the spatial location of cells, which is detrimental to studying the specific functions and intercellular communications of different cell populations in their spatial context [212,218]. In the future, combining scRNA-seq with spatial transcriptomics will further reveal the interactions between cells and their microenvironment.

Single-cell sequencing methods have been rapidly evolving in recent years, and multi-omic approaches can comprehensively analyze the molecular mechanisms of ovarian development in terms of the genome, epigenome, transcriptome, proteome, metabolism, and microenvironment. The application of scRNA-seq alone or in combination with other single-cell sequencing technologies will enable people to understand more clearly the physiological function and ultra-anatomy of ovarian tissue, the regularity of oocyte development, and the molecular network of the cell–cell interactions, which is beneficial for the diagnosis and treatment of ovarian-related diseases and the possibility of retarding the ovarian aging process.

## Figures and Tables

**Figure 1 biomolecules-13-00047-f001:**
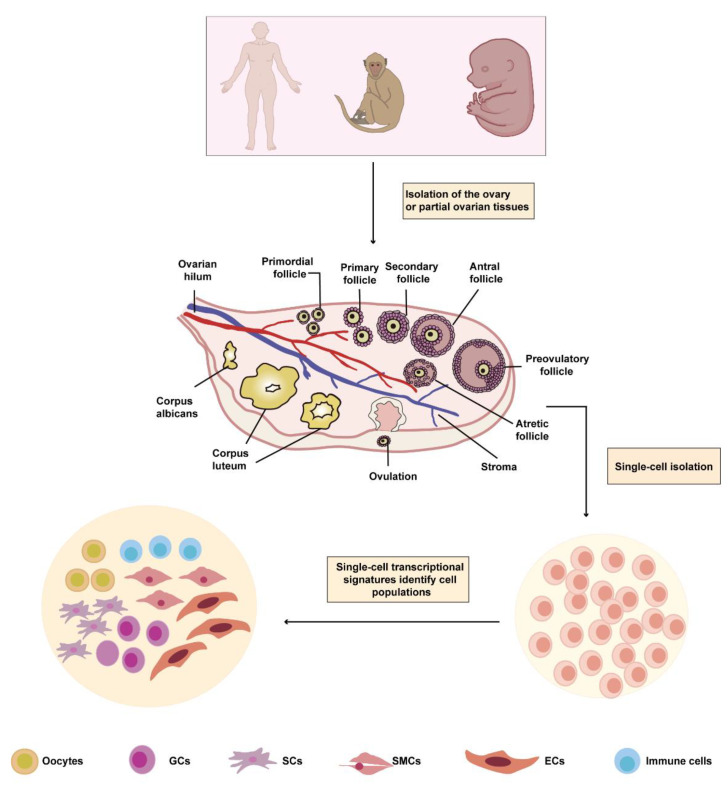
scRNA-seq technology identifies distinct cell subsets in the ovary of human, nonhuman primate, and mouse. The inner ovarian cortex of an adult human, the ovary of an adult human, human embryo, cynomolgus monkey, fetal mouse, and mouse embryo are collected. Then, the tissue is dissected to prepare a single-cell suspension, and cell subpopulations are identified based on the transcriptome data and specifically expressed genes. The ovary mainly contains six cell types, including oocytes, GCs, SCs, SMCs, ECs, and immune cells. The current scRNA-seq research has not found SMCs in the ovary of fetal mice ranging from 16.5 days postcoitum–PD3 or in those of human embryos between 5 and 26 weeks postfertilization. GCs, granulosa cells; SCs, stromal cells; SMCs, smooth muscle cells; ECs, endothelial cells.

**Figure 2 biomolecules-13-00047-f002:**
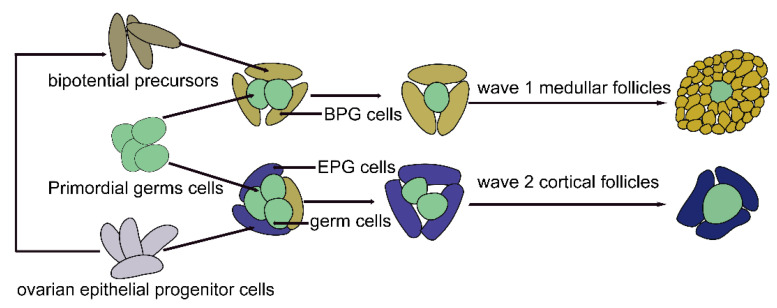
Schematic diagram of the origin and development of pregranulosa cells. BPG cells originate from bipotential precursors, which are derived from ovarian epithelial progenitor cells. BPG cells enclose several primordial germ cells to form germ cell nests. Next, the nest breaks down, and BPG cells differentiate into granulosa cells of wave 1 follicles, which began to develop immediately after birth in the medullar region. Additionally, some BPGs may remain in the cortex for some time. With the development of the embryo, cortical BPG cells are replaced by EPG cells, which do not significantly reside in the ovarian medulla. EPG cells are derived from ovarian epithelial progenitor cells and become the granulosa cells of wave 2 follicles in the ovarian cortex, which represent the ovarian reserve (BPG, bipotential pregranulosa; EPG, epithelial pregranulosa).

**Figure 3 biomolecules-13-00047-f003:**
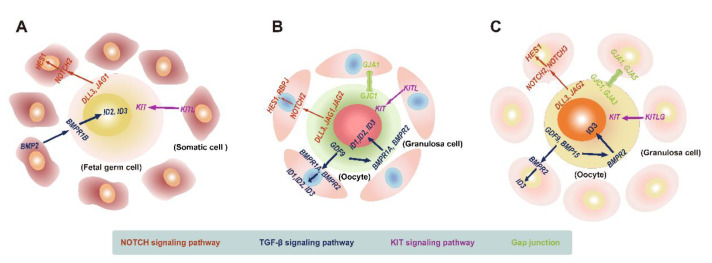
Canonical intercellular interactions in the ovary. (**A**) Schematic diagram showing the interaction of female human fetal germ cells and gonadal somatic cells mediated by typical signaling pathways. For the NOTCH signaling pathway, *DLL3* and *JAG1* is expressed on fetal germ cells, and the NOTCH receptor (*NOTCH2*) as well as targets (*HES1*) are expressed on somatic cells; for the TGF-β signaling pathway, *BMP2* is expressed on somatic cells, and its receptor (*BMPR1B*) and target genes are expressed on fetal germ cells; for the KIT signaling pathway, the ligand *KITL* is expressed on somatic cells, while the *KIT* receptor is expressed on fetal germ cells. (**B**) Schematic illustration displaying the bidirectional communication between oocytes and granulosa cells during the primordial follicular assembly. For the NOTCH signaling pathway, the ligands *DLL3, JAG1*, and *JAG2* are expressed on oocytes, and NOTCH receptors (*NOTCH2*) as well as targets (*HES1* and *RBPJ*) are expressed on granulosa cells; for the TGF-β signaling pathway, *GDF9* is expressed on oocytes, while its receptors (*BMPR1A* and *BMPR2*) and targets (*ID1, ID2*, and *ID3*) are expressed on oocytes and granulosa cells; for the KIT signaling pathway, the ligand *KITL* is expressed on granulosa cells, while the receptor *KIT* is expressed on oocytes; for gap junctions, *GJA1* is expressed on granulosa cells, and *GJC1* is expressed on oocytes. (**C**) Schematic diagram showing the characteristics of the intercellular crosstalk between oocytes and granulosa cells throughout folliculogenesis. For the NOTCH signaling pathway, the ligands *DLL3* and *JAG2* are expressed on oocytes, and NOTCH receptors (*NOTCH2* and *NOTCH3*) as well as targets (*HES1*) are expressed on granulosa cells; for the TGF-β signaling pathway, *GDF9* and *BMP15* are expressed on oocytes, and its receptor (*BMPR2*) and targets (*ID3*) are expressed on oocytes and granulosa cells; for the KIT signaling pathway, the *KITLG* ligand is expressed on granulosa cells, while the *KIT* receptor is expressed on oocytes; for gap junctions, *GJA1* and *GJA5* are expressed on granulosa cells, and *GJC1* and *GJA3* are expressed on oocytes.

**Table 1 biomolecules-13-00047-t001:** Possible marker genes of ovarian cells in different species demonstrated by single-cell RNA sequencing.

Cell Type		Marker Genes	Species	Technology	Reference
Fetal germ cells (FGCs)	FGCs	*DDX4, DAZL, POU5F1, PRDM1, TFAP2C, KIT, NANOG, SALL4, LIN28A, LEFTY1, LEFTY2*	Mouse	10XGenomics	[47]
		*DDX4, DAZL*	Mouse	10XGenomics	[48]
	Mitotic FGCs	*POU5F1, NANOG*	Human	Smart-seq2	[39]
	Retinoid-acid-signaling-responsive FGCs	*STRA8, ZGLP1, ANHX, ASB9, THRA/BTR*	Human	Smart-seq2	[39]
	Meiotic prophase FGCs	*Il13RA2*	Human	Smart-seq2	[39]
	Oogenesis stage FGCs	*PECAM1, ZP3, OOSP2*	Human	Smart-seq2	[39]
Oocyte		*GDF9, ZP3, FIGLA, OOSP2*	Human	10XGenomics	[34]
		*DDX4, ZP2, SYCP3, SOX30, ZAR1, DAZL*	Human	Tang method	[49]
		*DDX4, ZP2, FIGLA, SOX30*	Human	Smart-seq2	[50]
		*GDF9, DDX4, SYCP3, ZP3, LMOD3, RBM46 NET01*	Monkey	STRT-seq	[45]
		*DDX4, DAZL*	Mouse	10XGenomics	[36,38]
		*DAZL*	Mouse	10XGenomics	[37]
Pregranulosa cells		*FOXL2, WNT4, WNT6, KITL*	Mouse	10XGenomics	[48]
Granulosa cells (GCs)		*FOXL2, WT1*	Human	Smart-seq2	[39]
		*FOXL2, AMH*	Human	10XGenomics	[34]
		*AMH, CYP11A1, STAR, INHBA*	Human	10XGenomics	[49]
		*FOXL2, CDH2, GJA1, TNNI3*	Human	Smart-seq2	[50]
		*AMH, HSD17B1, SERPINE2, GSTA1*	Human	10XGenomics	[35]
		*AMH, WT1, INH4*	Monkey	STRT-seq	[45]
		*AMHR2, KITL*	Mouse	10XGenomics	[36,38]
		*AMHR2*	Mouse	10XGenomics	[37]
		*AMH, AMHR2, KITL, CYP19A1, FSHR*	Mouse	10XGenomics	[51]
	Cumulus GCs	*IGFBP2, INHBB, IHH*	Human	Smart-seq2	[50]
		*IGFBP2, INHBB, IHH, VCAN, FST, HTRA1*	Human	10XGenomics	[35]
		*HAS2, NPR2*	Mouse	10XGenomics	[51]
	Mural GCs	*KRT18, AKIRIN1, CYP19A1,*	Human	Smart-seq2	[50]
		*KRT18, AKIRIN1, LIHP, CITED2*	Human	10XGenomics	[35]
		*FSHR, BMPR2, NPPC*	Mouse	10XGenomics	[51]
Stromal cells		*DCN, COLLA1, COL6A1, PDGFRA*	Human	10XGenomics	[34]
		*DCN, COLLA1, COL3A1, APOE*	Human	Smart-seq2	[50]
		*DCN, LUM*	Human	10XGenomics	[35]
		*TCF21, COLLA2*	Monkey	STRT-seq	[45]
		*MFAP4, NR2F2*	Mouse	10XGenomics	[38]
		*TCF21, NR2F2*	Mouse	10XGenomics	[36]
		*MFAP4*	Mouse	10XGenomics	[37]
Smooth muscle cells		*TAGLN, RGS5, MYH11, MCAM, RERGl*	Human	10XGenomics	[34]
		*TAGLN, RGS5*	Human	10XGenomics, Smart-seq2	[35,50]
		*DES, ACTA2*	Monkey	STRT-seq	[45]
Endothelial cells		*CDH5 (CD144), PECAM1 (CD31)*	Human	Smart-seq2	[39]
		*vWF, CDH5*	Human	10XGenomics	[34]
		*vWF,* * CD34 *	Human	10XGenomics	[50]
		*vWF, CLDN5*	Human	10XGenomics	[35]
		*vWF, CDH5*	Monkey	STRT-seq	[45]
		*APLNR, EGFL7*	Mouse	10XGenomics	[38]
		*APLNR*	Mouse	10XGenomics	[36,37]
Immune cells	Immune cells	*CD69, ITGB2*	Human	10XGenomics	[34]
		*CD53, CXCR4*	Human	10XGenomics	[35]
		*ELNE, MPO*	Mouse	10XGenomics	[38]
		*TYROBP*	Mouse	10XGenomics	[36,37]
	Innate immune cells	*CD68, IFI30*	Human	10XGenomics	[35]
	NK cells	*CD3d, KLRB1*	Monkey	STRT-seq	[45]
	Macrophages	*CD68, CD14*	Monkey	STRT-seq	[45]
	Antigen-presenting cells	*CD14, HLA-DRA, B2M, HLA-DQB1*	Human	10XGenomics	[34]
	T lymphocytes	*CD2, CD3G, CD8A*	Human	10XGenomics	[34]
	T lymphocytes	*AW112010, CD3G*	Mouse	10XGenomics	[51]
	B lymphocytes	*IGHM, CD37*	Mouse	10XGenomics	[51]
	Monocytes or monocyte-derived cells	*CD14*	Mouse	10XGenomics	[51]

Smart-seq, switching mechanism at 5′ end of the RNA transcript; STRT-seq, single-cell tagged reverse transcription sequencing.

## Data Availability

Not applicable.

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
