# Peer review of "Application of Single-Cell RNA Sequencing in Ovarian Development"

_biomolecules, 2022, doi:10.3390/biom13010047_

Round 1

Reviewer 1 Report

In this manuscript, the authors summarize the findings of ovarain cells revealed by single cell RNA-seq. The marker genes of cellular subpopulations, the important devolopmental  processes and the key signaling pathways for CCI in the ovary of humans, nonhuman primates and mice were reviewed thoroughly based on the scRNA-seq study in the past several years.  

Specific comments are detailed below:

1. There are several cell tpyes  shared by ovary and other tissues, such as stromal cells, endothelial cells, SMC and immune cells. Are there some ovarian specific marker genes in these cell types?   If there are, what are the biological function of the specific marker genes?

2. The authors claim the review is benifical to treatment of ovrain-related diseases.  But there is no scRNA-seq study in the review which include the single cell data of ovrain-related diseases. If possible, the authors may add some diseases-related scRNA-seq study.

3. In table 1, the authos list the possible marker genes of ovarian cells, the marker genes of the same cell type from differnet study are slightly different. Is it because of the different scRNA-seq technologies or other factors? Can the authors provide some explainations? 

Author Response

The revised manuscript can be seen in the attachment.

Response to Reviewer 1 Comments

Point 1: There are several cell tpyes shared by ovary and other tissues, such as stromal cells, endothelial cells, SMC and immune cells. Are there some ovarian specific marker genes in these cell types?   If there are, what are the biological function of the specific marker genes?

Response 1: Thank you for your comment. You have asked an interesting question. We have carefully reviewed all single-cell sequencing articles about ovary that have been cited in this manuscript. But we found there are no ovarian-specific marker genes in ovarian stromal cells, endothelial cells, SMC, and immune cells. Surprisingly, some cells in the ovarian stromal cells expressed high levels of steroidogenic acute regulatory (STAR). STAR plays a key role in steroidogenesis, which accelerates the transport of cholesterol to the inner mitochondrial membrane where the cytochrome P450scc enzyme is located. It may be important for ovarian follicle development and maturation.

Point 2: The authors claim the review is benifical to treatment of ovrain-related diseases.  But there is no scRNA-seq study in the review which include the single cell data of ovrain-related diseases. If possible, the authors may add some diseases-related scRNA-seq study.

Response 2: Thank you for your valuable advice. We have added some scRNA-seq studies about ovarian-related diseases, such as polycystic ovarian syndrome and ovarian cancer. They can be found in the following paragraphs:

  • paragraphs 253 to 255: Oocyte maturation can be affected by some ovarian-related diseases (for example, polycystic ovarian syndrome (PCOS)) [80].
  • paragraphs 371 to 383: 3.2.4 The characteristics of oocytes from polycystic ovarian syndrome patients. The oocytes in PCOS patients are often of poor quality, leading to lower fertilization, cleavage, and implantation rates [108]. It is essential to elucidate the mechanism behind this in detail. By analyzing the scRNA-seq data from 14 oocytes from 7 healthy fertile women and 20 oocytes from 9 patients with PCOS at the germinal vesicle (GV) stage, metaphase I stage, and metaphase II (MII) stage, Qi et al. discovered that some genes associated with mitochondrial function (for example, oxidative phosphorylation), such as COX6B1, COX8A, COX4l1, and NDUFB9 were prematurely activated at the GV stage of PCOS oocytes, whereas it occurs at the MII stage in healthy oocytes [109]. Mitochondrial is essential for the oocyte maturation, meiotic spindle assembly, fertilization, and subsequent preimplantation embryogenesis [110]. Abnormal function of the mitochondria may account for the low-quality oocytes in PCOS patients, which provides a new idea for improving the reproductive outcome of PCOS patients.)
  • paragraphs 611 to 614: Furthermore, stromal cells obtained from primary ovarian tumors were enriched for upregulated extracellular matrix genes and genes associated with the epithelial-to-mesenchymal transition (EMT) [167]. EMT plays an important role in tumor cell metastasis. This characteristic is associated with poor ovarian cancer prognosis [168].

Point 3: In table 1, the authos list the possible marker genes of ovarian cells, the marker genes of the same cell type from differnet study are slightly different. Is it because of the different scRNA-seq technologies or other factors? Can the authors provide some explainations?

Response 3: Thank you for your comments. You have raised an important question. We also find it very confusing. So we summarized the scRNA-seq techniques used in the different articles according to your suggestions. This information has been added to table1. But it could not fully explain the phenomenon. Besides, we discovered there were two different ways to identify cell types. One method is manual cell annotation. The cells are clustered using an unsupervised method, and then the clusters are annotated to different cell types based on canonical markers found in the differentially expressed genes of the cluster. Another method is automatic cell type identification. It does not require manual annotation. Instead, they can be used to predict the cell types directly from the public resources of scRNA-seq data. The input of eager learning and lazy learning methods are the training datasets and testing datasets. The input of marker learning methods is the markers of each cell type and the testing datasets. The training datasets can be downloaded from the data resource centers (GEO, ArrayExpress and GSA). The markers of each cell type can be downloaded from the marker resource centers (PanglaoDB, CellMarker and CancerSEA). The methods used by eager learning, lazy learning, and marker learning methods are classifiers, nearest neighbor cells, and scoring functions, respectively. The cell types assigned by the automatic methods can be given to cells or clusters. But this method can get many marker genes in the same cell type of the same tissue. Researchers may choose to place different contents according to their research purpose. We think it may also be one reason that the marker genes of the same cell type from different studies are slightly different.

Reviewer 2 Report

The review details the information on the application of single-cell RNA to different types of ovarian cells. The review is well written and is very extensive. In general there are few reviews with this topic and it provides a lot of relevant information. In paragraphs 722 to 732 there are different font sizes as well as different formatting (italics). 

Author Response

The revised manuscript can be seen in the attachment.

Response to Reviewer 2 Comments

Point 1: The review details the information on the application of single-cell RNA to different types of ovarian cells. The review is well written and is very extensive. In general there are few reviews with this topic and it provides a lot of relevant information. In paragraphs 722 to 732 there are different font sizes as well as different formatting (italics). 

Response 1: Thank you for your valuable advice. We have changed the different font sizes as well as different formatting according to your suggestion (The NOTCH signaling ligands DLL3, JAG1, and JAG2 are mainly expressed by oocytes, whereas the NOTCH2 receptor is expressed by GCs. The target genes HES1 and RBPJ were expressed in both oocytes and GCs, although the expression of HES1 and RBPJ in oocytes was transient. For the TGF-β signaling pathway, its important component GDF9 exists in oocytes, while its receptors BMPR1A and BMPR2, as well as its target genes ID1, ID2, and ID3 were found both in oocytes and GCs, implying that the PF formation is regulated by autocrine and paracrine effects.). Besides, we also carefully checked the manuscript. And we changed the different font sizes as well as different formatting in paragraphs 596 to 598 (Because theca cells are generated from stromal cells [164], they may have the same cell markers. In addition, mouse stromal cells specifically express TCF21 and NR2F2 [36].), and paragraphs 749 to 751(Except for those classic pathways, gap junctions also participate in the interactions between oocytes and granulosa cells during the formation of PFs, with oocytes expressing GJC1 and GCs expressing GJA1 [36, 196] (Figure 3B).).

Reviewer 3 Report

It was a great pleasure to review the manuscript entitled "Application of Single-cell RNA Sequencing in Ovarian Development " submitted for consideration in Biomolecules. The review article is well-organized and informative. One major part missing in the article is using Single-cell RNA Sequencing to detect ovarian cancer. The other suggestion I have is on the reference section. Some references are either missing volume numbers or the year was repeated as volume. Ex: Ref no. 5, 7, 20, etc.

Author Response

The revised manuscript can be seen in the attachment.

Response to Reviewer 3 Comments

Point 1: One major part missing in the article is using Single-cell RNA Sequencing to detect ovarian cancer.

Response 1: Thank you for your precious suggestion. We have added some information to detect ovarian cancer with scRNA-seq. Besides, we also added some information to detect oocytes from polycystic ovarian syndrome patients with scRNA-seq. They can be found in the following paragraphs:

  • paragraphs 253 to 255: Oocyte maturation can be affected by some ovarian-related diseases (for example, polycystic ovarian syndrome (PCOS)) [80].
  • paragraphs 371 to 383: 3.2.4 The characteristics of oocytes from polycystic ovarian syndrome patients. The oocytes in PCOS patients are often of poor quality, leading to lower fertilization, cleavage, and implantation rates [108]. It is essential to elucidate the mechanism behind this in detail. By analyzing the scRNA-seq data from 14 oocytes from 7 healthy fertile women and 20 oocytes from 9 patients with PCOS at the germinal vesicle (GV) stage, metaphase I stage, and metaphase II (MII) stage, Qi et al. discovered that some genes associated with mitochondrial function (for example, oxidative phosphorylation), such as COX6B1, COX8A, COX4l1, and NDUFB9 were prematurely activated at the GV stage of PCOS oocytes, whereas it occurs at the MII stage in healthy oocytes [109]. Mitochondrial is essential for oocyte maturation, meiotic spindle assembly, fertilization, and subsequent preimplantation embryogenesis [110]. Abnormal function of the mitochondria may account for the low-quality oocytes in PCOS patients, which provides a new idea for improving the reproductive outcome of PCOS patients.)
  • paragraphs 611 to 614: Furthermore, stromal cells obtained from primary ovarian tumors were enriched for upregulated extracellular matrix genes and genes associated with the epithelial-to-mesenchymal transition (EMT) [167]. EMT plays an important role in tumor cell metastasis. This characteristic is associated with poor ovarian cancer prognosis [168].

Point 2: The other suggestion I have is on the reference section. Some references are either missing volume numbers or the year was repeated as volume. Ex: Ref no. 5, 7, 20, etc.

Response 2: Thank you for your careful work. We have carefully checked the reference section. We also have checked the original article. We found the volume number did not miss and the year was not repeated. The reason is following: take Ref no. 5 as an example: He, J.; Yao, G.; He, Q.; Zhang, T.; Fan, H.; Bai, Y.; Zhang, J.; Yang, G.; Xu, Z.; Hu, J.; Sun, Y., Theaflavin 3, 3'-Digallate Delays Ovarian Aging by Improving Oocyte Quality and Regulating Granulosa Cell Function. Oxidative medicine and cellular longevity 2021, 2021, 7064179. The second 2021 represents the real volume rather than a repeated year. And this article does not have volume numbers. 7064179 represents Article ID. And all references are inserted via endnote.